# Identification of Potential Surface Water Resources for Inland Aquaculture from Sentinel-2 Images of the Rwenzori Region of Uganda

**Athanasius Ssekyanzi** [1,2,*] **, Nancy Nevejan** [1] **, Dimitry Van der Zande** [3] **, Molly E. Brown** [4] **and Gilbert Van Stappen** [1]

1 Laboratory of Aquaculture & Artemia Reference Center, Department of Animal Sciences and Aquatic Ecology, Ghent University, Coupure Links 653, 9000 Gent, Belgium; Nancy.Nevejan@UGent.be (N.N.); Gilbert.VanStappen@UGent.be (G.V.S.)
2 School of Agriculture and Environmental Sciences, Mountains of the Moon University, Fort Portal P.O. Box 367, Uganda
3 Operational Directorate Natural Environment, Royal Belgian Institute of Natural Sciences (RBINS), 1000 Brussels, Belgium; dvanderzande@naturalsciences.be
4 Natural Resources Institute, University of Greenwich, Central Avenue, Chatham Maritime, Kent ME4 4TB, UK; M.E.Brown@greenwich.ac.uk
* Correspondence: ssekyanziarthur@gmail.com or Athanasius.Ssekyanzi@UGent.be; Tel.: +256-773-962-035

**Abstract:** Aquaculture has the potential to sustainably meet the growing demand for animal protein. The availability of water is essential for aquaculture development, but there is no knowledge about the potential inland water resources of the Rwenzori region of Uganda. Though remote sensing is popularly utilized during studies involving various aspects of surface water, it has never been employed in mapping inland water bodies of Uganda. In this study, we assessed the efficiency of seven remote-sensing derived water index methods to map the available surface water resources in the Rwenzori region using moderate resolution Sentinel 2A/B imagery. From the four targeted sites, the Automated Water Extraction Index for urban areas (AWEInsh) and shadow removal (AWEIsh) were the best at identifying inland water bodies in the region. Both AWEIsh and AWEInsh consistently had the highest overall accuracy (OA) and kappa (OA > 90%, kappa > 0.8 in sites 1 and 2; OA > 84.9%, kappa > 0.61 in sites 3 and 4), as well as the lowest omission errors in all sites. AWEI was able to suppress classification noise from shadows and other non-water dark surfaces. However, none of the seven water indices used during this study was able to efficiently extract narrow water bodies such as streams. This was due to a combination of factors like the presence of terrain shadows, a dense vegetation cover, and the image resolution. Nonetheless, AWEI can efficiently identify other surface water resources such as crater lakes and rivers/streams that are potentially suitable for aquaculture from moderate resolution Sentinel 2A/B imagery.

**Keywords:** water; water index; optimum threshold; omission error; NDWI; MNDWI1; MNDWI2; AWEIsh; AWEInsh; MuWI_C; MuWI_R; Sentinel-2; aquaculture; Rwenzori region

## 1. Introduction

Uganda is a landlocked country, with about 16% (37,166 km$^2$) of its land area covered by surface water bodies and wetlands [1,2]. This presents an opportunity to satisfy the growing demand for fish through aquaculture, and so contribute to assuring healthy diets for Uganda's growing population as aligned to the targets of Sustainable Development Goal 2 [3]. On the contrary, malnutrition is still a major public health problem in the country [4]. For example, at least 41% of the children between 6 and 59 months in the Rwenzori region are undersized [5–7]. Improved aquaculture production could be a reliable pathway toward meeting the protein, micronutrient, and essential fatty acids needs of vulnerable populations in this region. The Rwenzori region has an abundance of natural

water bodies with good water quality [8–10] that can be used for fish farming, yet there are no specific plans of utilizing them for aquaculture development.

Remote sensing is a proven technique in mapping water bodies [11–19], assessing floods [15,20–25], water quality [15,22–27], and estimating water scarcity [15,22–25]. The Sentinel-2 satellite, launched by the European Space Agency (ESA) in 2015, has played a key role in providing multispectral images for water body mapping [13]. Sentinel-2 images have a moderate spatial resolution (10 m, 20 m, and 60 m) suitable for regional water body mapping, due to their free access coupled with the frequent revisit capabilities [13] of 10 days for the studied region. Compared to the Landsat series, Sentinel-2 delivers images with higher spatial resolution, more spectral bands, more frequent revisit time, and wider swath, thus showing great potential in aquatic science studies [15,28–32].

Several other lower spatial resolution sensors such as Terra/Aqua Moderate Resolution Imaging Spectroradiometer (MODIS), Sentinel-3, TOPEX/Poseidon, Jason-1, Jason-2, Jason-3, and Envisat also provide open access satellite optical imagery [33–37] but their use in water body mapping studies is limited to larger inland lakes and reservoirs [37]. Surface Water and Ocean Topography (SWOT) is another optical sensor due to be launched in 2022 [36] but its application will be limited to wide rivers (100 m or wider) and lakes with a surface area of at least 250 m $\times$ 250 m) [36,37]. Ultra-high spatial resolution sensors such as Pleiades, IKONOS, QuickBird, WorldView, RapidEye, ZY-3, and GF-1/GF-2 [38–40] do exist but their data are not open access, which makes their use expensive.

The water index method is the most commonly employed when extracting water bodies from remotely sensed data [41]. Compared to other common methods such as pixel-based and object-based classification, index-based methods extract pure and mixed water pixels better in challenging environments [42]. Water index methods include a variety of techniques such as Normalized Difference Water Index (NDWI) [12,14,41,43], Modified Normalized Difference Water Index (MNDWI) [44], and the Automated Water Extraction Index (AWEI) [16].

NDWI uses the green and near-infrared (NIR) bands to identify water from other land features though it cannot easily differentiate water from built-up areas such as asphalt roads and highly reflective house roofs [41,43]. The near-infrared (NIR) band utilized in NDWI was replaced with the shortwave infrared (SWIR) band by Xu (2006) [44] to derive the Modified Normalized Difference Water Index (MNDWI). This was done to resolve the issues of NDWI [43,44]. Nevertheless, MNDWI still possesses a major problem of misclassifying shadows in mountainous terrains [43].

The Automated Water Extraction Index (AWEI) was proposed by Feyisa et al. (2014) [16]. AWEI improves the precision of surface water mapping by suppressing classification noise from shadows and other non-water dark surfaces [16]. This water index constitutes two versions: AWEIsh that removes shadow pixels, and AWEInsh that is recommended for urban regions [16,43]. The latter was specifically formulated to eliminate non-watery pixels of dark built-up surfaces in areas with an urban setting, while the former removes shadow pixels that AWEInsh cannot eradicate [16].

Recently, the new multi-spectral water index (MuWI) which consists of a complete version (MuWI_C) and the revised one (MuWI_R) [15] has gained prominence. MuWI_C contains many terms that are considered redundant to water mapping, making it long and complicated [15]. Therefore, it was refined using four highly weighted terms with integer coefficients to derive the shorter and simpler MuWI_R [15]. These multi-spectral water indices combine several normalized differences of bands to produce high resolution (10 m) water maps without band sharpening, while also ensuring a stable threshold [15]. They use the support vector machine (SVM) machine-learning algorithm [15]. MuWI increases spatial resolution and lowers the commission as well as omission errors [15]. This is attributed to its efficiency in exploiting the native 10-m Sentinel-2 spectral bands, plus the significant reductions in shadow and sunglint misclassifications [15].

These water index methods require the implementation of thresholds that enable the separation of water from environmental noises such as shadows, forests, built-up areas,

snow, and clouds [43,45]. The considered bands are comprehensively examined [14,46] to determine the threshold that categorizes water from confusing non-water bodies [14,44]. Thresholding makes the extraction of water bodies simpler and faster [43]. Although standard thresholds exist, they were inefficient in another challenging environment of Nepal according to the results of Acharya et al. (2018) [43]. Their study employed a trial and error method in identifying the optimum threshold [43]. Here we use k-means cluster analysis to obtain the optimum threshold. This brings the advantages of lessening the number of calculations as well as eliminating the errors caused by the artificial selection of empirical values [47,48].

In this study, the efficiency of various water indices (NDWI, MNDWI1, MNDWI2, AWEIsh, AWEInsh, MuWI_C, and MuWI_R) at mapping the available water resources (narrow streams, rivers, lakes, swamps) for potential inland freshwater aquaculture in the Rwenzori region was assessed. This study is the first of its kind in Uganda and the obtained results will be utilized in further research of identifying potential sites for inland aquaculture in the Rwenzori region of Uganda.

## 2. Materials and Methods

### 2.1. Methodology Workflow

The workflow followed in this study included image acquisition, image pre-processing, derivation of water indices, k-means cluster analysis, determination of the optimum threshold, and binary classification of water and non-water (Figure 1). All the processing was done in QGIS version 3.14.16 "Pi", SNAP 7.0.0, and R 3.6.1 (The R Foundation, Vienna, Austria) software packages.

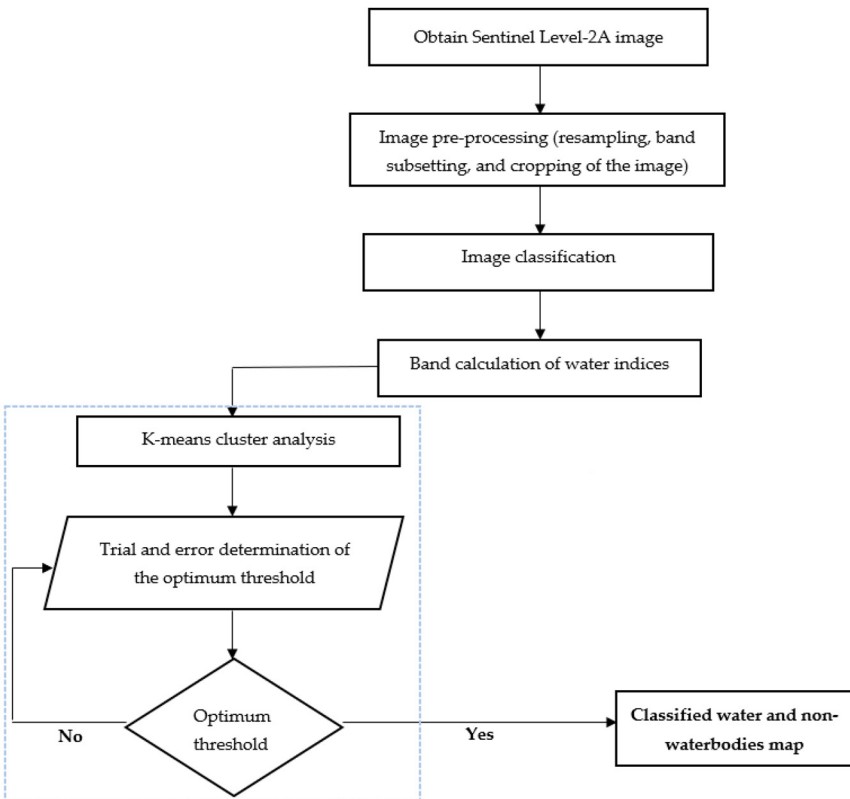

**Figure 1.** Workflow of the image classification exercise. The blue rectangle highlights the activities involved in identifying the optimum thresholds for the various water indices. Index values obtained from band calculations were grouped using k-means cluster analysis. The cluster centers were compared for accuracy using a trial and error method The cluster centroid(s) that gave the highest overall accuracy and kappa was selected as the optimum threshold for that particular water index. These were then employed in the binary classification of water and non-water.

*2.2. Study Area*

The Rwenzori region is estimated to cover an area of 7500 km$^2$ (approximately 3.1% of the country) that is close to the equator along the border of Uganda and the Democratic Republic of Congo (DRC) [49]. The region is bounded by the Kazinga channel, Lakes George and Edward to the south, protected areas to the east, Lake Albert to the north, and DRC and Rwenzori mountain ranges to the west [49]. Among the nine districts of the Rwenzori region, the study area in this exercise covered four sites from three districts (Kasese, Kabarole, and Ntoroko, Figure 2). The rugged terrain of this region is characterized by the presence of several permanent wetlands, streams/rivers, ponds, and crater lakes of varying sizes. Site 1 covers portions of both the Ntoroko and Kabarole districts. The area has an elevation ranging between 1290 and 1579 m above sea level (masl). Site 2 covers a portion of the southern part of the Kabarole district that has an elevation ranging between 950 and 1579 masl. Both these sites (1 and 2) have several crater lakes, narrow streams, rivers, and swampy areas. Sites 3 and 4 are both located in Kasese district, which is a drier region. The elevation of the areas covered by these two sites ranges between 950 and 1579 masl. Site 3 is characterized by River Mobuku and its flood plains, dispersed tiny urban centers, as well as a large swampy area to its southwest. Site 4 covers a portion of lake George, swampy-vegetated areas, crater lakes, River Nyamusagani to the North-west, and several patches of bare ground and urban centers (Figure 2).

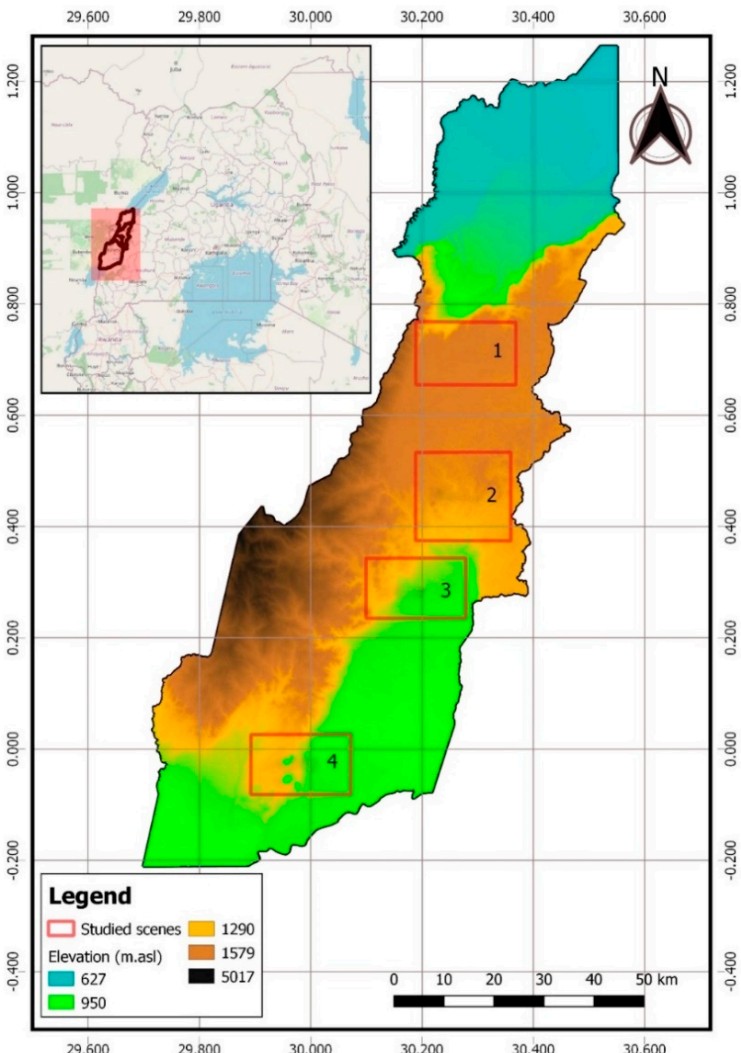

**Figure 2.** Map of the Rwenzori region of Uganda along with the elevation range. The red rectangles represent the studied sites (1, 2, 3, and 4).

*2.3. Image Acquisition and Pre-Processing*

A Sentinel-2 Level 2A (L2A) product was downloaded from ESA Sentinel-2 Open Access Hub (https://scihub.copernicus.eu/ (accessed on 13 April 2020)). L2A products provide a Bottom of Atmosphere (BOA) reflectance [50]. L2A products are obtained by atmospherically correcting Sentinel-2 MSI level-1C products using the Sen2Cor method [51,52]. The only shortfall is that the processor does not consider water surface effects such as sunglint [52]. Surface reflectance (or BOA reflectance) rather than Top of the Atmosphere (or TOA) reflectance, theoretically delineates surface water more accurately because it removes atmospheric disturbances [15]. The image was obtained by the satellite on 28 January 2019 at 08 12 h over tile 35NRA during a relative orbit of 078. It was processed with the Payload Data Ground Segment (PDGS) Processing Baseline at 10 55 h. The obtained image had cloud cover, cloud shadow, and dark features percentages of 1.89, 0.34, and 0.36 respectively, basing on the basic L2 metadata. To ensure that all the bands had the same resolution (10 m), resampling with the nearest neighbor method [53] was carried out in SNAP 7.0.0 software using band 2 as the reference band. The various bands required for the derivation of water indices were subsetted into a single raster stack using SNAP 7.0.0 software. The sites to be investigated for the presence of surface water resources were cropped from the stacked raster using R 3.6.1 software. Except for site 2 that was cropped to a size of 1900 by 1750 pixels to cover all the crater lakes in that region of Kabarole district, the other three scenes (1, 3, and 4) were cropped to a uniform size of 2000 by 1200 pixels. RGB images of the studied sites are shown in Figure 3.

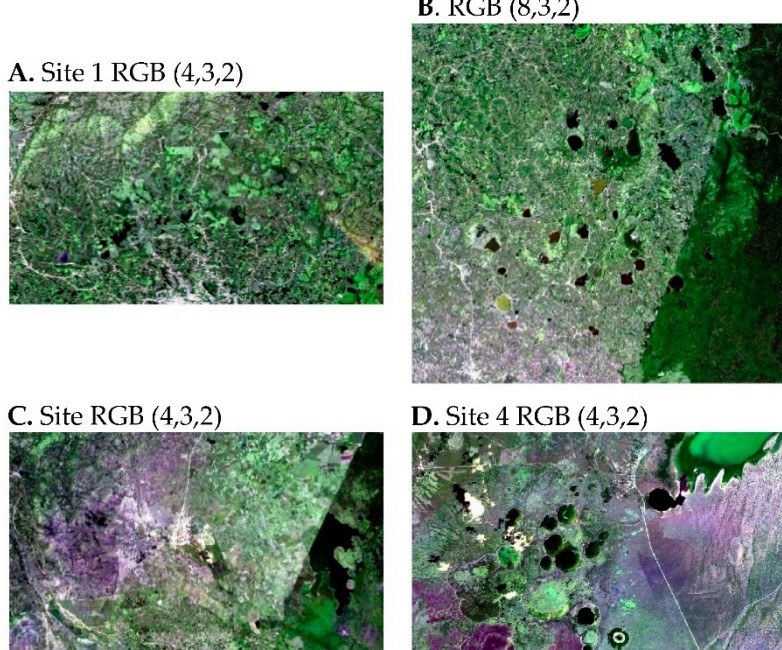

**Figure 3.** Sentinel-2 MSI RGB (4,3,2) 10 m natural-color images of the studied sites. Frames (**A**–**D**) are studied sites 1, 2, 3, and 4, respectively.

*2.4. Derivation of Water Indices*

Derivation of water indices for binary classification of the water and non-water background was done using band mathematics equations using R 3.6.1 software (Table 1).

**Table 1.** Spectral water indices and their derivative equations. Bi refers to the particular Sentinel-2 band i; ND(i, j) denotes the normalized difference of Sentinel-2 band i and band j.

| Index | Equation | References |
|---|---|---|
| NDWI | (B3 − B8)/(B3 + B8) | [12,14,15,41,43,47] |
| MNDWI1 | (B3 − B11)/(B3 + B11) | [13,15,43,44,47] |
| MNDWI2 | (B3 − B12)/(B3 + B12) | [16,43] |
| AWEIsh | B2 + 2.5 × B3 − 1.5 × (B8 + B11) − 0.25 × B12 | [16,43] |
| AWEInsh | 4 × (B12− B11) − (0.25 × B8 + 2.75 × B11) | [16,43] |
| MuWI_C | −16.4ND(B2, B3) − 6.9ND(B2, B4) − 8.2ND(B2, B8) − 8.8ND(B2, B11) + 9.6ND(B2, B12) + 10.8ND(B3, B8) + 6.1ND(B3, B11) + 13.6ND(B3, B12) − 0.28ND(B4, B8) − 3.9ND(B4, B11) − 2.1ND(B4, B12) − 5.3ND(B8, B11) − 5.3ND(B8, B12) − 5.3ND(B11, B12) − 0.33 | [15,47] |
| MuWI_R | −4ND(B2, B3) + 2ND(B3, B8) + 2ND(B3,B12) − ND(B3, B11) | [15,47] |

## 2.5. Derivation of the Optimum Threshold and Binary Classification

According to the findings of Acharya et al. (2018) [43], the standard thresholds were found to be inefficient at deriving surface water. Several challenges have been faced when using standard threshold values to separate water from the non-water background in regions that are characterized by hills, shades, forests, and urban areas [42,43,54]. Therefore, the standard thresholds were not utilized during the water extraction exercises in this study because the Rwenzori region has a majorly forested rugged terrain with sparse built-up areas. Rather, *k*-means clustering analysis [55] was employed to obtain the optimum threshold for each index. The k-means method is a commonly utilized algorithm for geometric clustering, which is also known as Lloyd's algorithm [56].

After deriving a particular water index for a given scene using band mathematics equations (Table 1), the index values were clustered into groups using *k*-means cluster analysis. The selected initial clusters (*k)* were 10. These were computed from a maximum of 500 iterations. The cluster-centers were compared for accuracy using a trial and error method [43]. The cluster centroids that gave the highest overall accuracy and kappa were selected as the optimum threshold for that water index. These procedures were carried out in R 3.6.1 software package.

The returned *k* cluster-centers for AWEI are illustrated in Table 2. The returned *k* cluster-centers for all the indices are shown in Table S5 of the Supplementary Materials.

**Table 2.** The returned *k* cluster centers for AWEI as derived from k-means clustering analysis. The highlighted cluster centroid values fell in the color range of water on each index plot and had the highest overall accuracy as well as kappa, and thus selected as the optimum thresholds of the respective water indices for that site.

| | AWEIsh | | | | AWEInsh | | |
|---|---|---|---|---|---|---|---|
| **Site 1** | **Site 2** | **Site 3** | **Site 4** | **Site 1** | **Site 2** | **Site 3** | **Site 4** |
| −5926.58 | −2928.46 | −2665.28 | −127.87 | −16,209.74 | −11,783.30 | −16,286.55 | −6280.94 |
| −3166.10 | −5199.38 | −5022.17 | −1936.92 | −11,294.86 | −270.36 | −18,559.13 | −953.94 |
| −4349.30 | −6411.62 | −4593.32 | −4496.02 | −12,176.02 | −8336.82 | −12,848.25 | −12,050.34 |
| −5493.35 | −4740.54 | −5394.81 | −4835.96 | −8619.31 | −13,910.72 | −13,819.54 | −15,416.24 |
| −5000.62 | −5596.51 | 881.26 | 1525.60 | −14,344.24 | −10,132.42 | −9000.94 | −14,205.73 |
| −6839.84 | 276.99 | −6182.35 | −5178.02 | −13,159.25 | −7201.59 | 876.09 | −9030.85 |
| 73.44 | −5986.37 | −7837.49 | −4069.98 | −7050.44 | −12,700.91 | −10,494.63 | −17,050.71 |
| −8286.82 | −4140.40 | −5761.15 | −6299.19 | −9596.69 | −10,951.91 | −11,801.94 | −13,076.04 |
| −6359.41 | −7658.28 | −3949.14 | −5562.80 | −10,458.41 | −15,888.88 | −14,897.65 | −10,825.84 |
| −7410.16 | −6929.70 | −6758.48 | −3141.01 | −444.37 | −9276.21 | −7571.75 | 2183.85 |

*2.6. Accuracy Assessment*

A confusion matrix-based approach [43,57] was used for assessing the accuracy of the binary classified water maps. A total of 800 reference points in each of the studied sites were used for this procedure (Figure S6). First, 500 random points were generated over each site area using the stratified random sampling method [58,59] in R 3.6.1 (The R Foundation, Vienna, Austria) software packages. These were manually labeled as water and non-water using high-resolution images (3 m) available from Google Earth Pro™ (Google Inc., Menlo Park, CA, USA) in QGIS version 3.14.16 "Pi". To these, 300 additional points that comprised a combination of confusing water and noisy non-water points (asphalt, built-up areas, dense vegetation, and bare ground) were added. These were based on authentication from the field and expert's knowledge of the area for small water bodies, as well as narrow river points. The procedures followed were similar to what was described by Acharya et al. (2018) [43].

During the comparisons of the binary classified water maps with reference datasets, the outcomes were four types of pixels:

- True-positive (TP): Number of correctly extracted water pixels.
- False-negative (FN): Number of undetected water pixels.
- False-positive (FP): Number of incorrectly extracted water pixels.
- True-negative (TN): Number of correctly rejected non-water pixels.
- Total (T): The total number of pixels in the accuracy assessment.

The outcomes of the confusion matrix were used to calculate the producer's accuracy (PA), user's accuracy (UA), overall accuracy (OA), kappa, omission (OE), and commission errors (CE) [12,15,43]. These were used to assess the accuracy of the produced maps from different water indices using R 3.6.1 (The R Foundation, Vienna, Austria) software packages.

## 3. Results

Results on the performance of the various water indices for the surface water extraction of four sites were assessed. From the equations illustrated in Table 1, index maps for each of the scenes were derived through binary classification using the optimum threshold. Quantitative accuracy assessment (Table 3) was carried out by comparing with high resolution (3 m) Google Earth images for reference. Confusion matrices for the whole study area are as shown in Table S7.

*3.1. Qualitative Analysis*

The produced water maps of the four sites after applying the optimum thresholds (highlighted values in Table 2 and Table S5) were visually assessed. Though the employed water index methods showed varying degrees of capability at extracting the crater lakes, lakes, and rivers/streams from all the studied sites, none of them was found to be completely efficient in this exercise.

NDWI, MNDWI1, and MNDWI2 misclassified built-up areas, dense vegetation, shadows of the terrain, bare ground, clouds, and clouds shadows as water (Figures S1–S4 of the Supplementary Materials). These water indices omitted whole or big portions of some water bodies and narrow streams/rivers. MuWI also misclassified shadows and vegetated areas as water. Visual analysis of the water maps produced using this index showed that even large portions of quite wide water bodies were omitted (Figure 4). This was most prevalent near the shorelines of these water bodies. On the contrary, MuWI indices did not misclassify high albedo surfaces to be water.

**Table 3.** Accuracy assessment of water maps for sites 1, 2, 3, and 4 of the different water indices based on the reference datasets. The OA and kappa of the best-performing indices for each site are highlighted.

| | NDWI | MNDWI1 | MNDWI2 | AWEIsh | AWEInsh | MuWI_C | MuWI_R |
|---|---|---|---|---|---|---|---|
| **Site 1** | | | | | | | |
| **PA (%)** | 44.00 | 61.30 | 61.30 | 84.20 | 84.20 | 59.80 | 69.90 |
| **UA (%)** | 95.90 | 92.10 | 93.10 | 94.90 | 93.70 | 99.40 | 97.40 |
| **OA (%)** | 80.80 | 85.40 | 85.60 | 93.20 | 92.90 | 86.50 | 89.40 |
| **OE (%)** | 56.02 | 38.72 | 38.72 | 15.79 | 15.79 | 40.23 | 30.08 |
| **CE (%)** | 4.10 | 7.91 | 6.86 | 5.08 | 6.28 | 0.63 | 2.62 |
| **Kappa** | 0.50 | 0.64 | 0.65 | 0.84 | 0.84 | 0.66 | 0.74 |
| **Site 2** | | | | | | | |
| | NDWI | MNDWI1 | MNDWI2 | AWEISH | AWEInSH | MuWI_C | MuWI_R |
| **PA (%)** | 30.50 | 42.00 | 53.20 | 78.10 | 81.40 | 53.90 | 57.00 |
| **UA (%)** | 91.10 | 98.30 | 87.20 | 95.50 | 98.60 | 99.30 | 98.70 |
| **OA (%)** | 75.40 | 80.10 | 81.50 | 91.30 | 93.30 | 84.20 | 85.40 |
| **OE (%)** | 69.52 | 57.99 | 46.84 | 21.93 | 18.59 | 46.10 | 42.38 |
| **CE (%)** | 8.89 | 1.74 | 12.80 | 4.55 | 1.35 | 0.68 | 1.27 |
| **Kappa** | 0.35 | 0.48 | 0.54 | 0.80 | 0.84 | 0.60 | 0.64 |
| **Site 3** | | | | | | | |
| | NDWI | MNDWI1 | MNDWI2 | AWEIsh | AWEInsh | MuWI_C | MuWI_R |
| **PA (%)** | 73.90 | 43.70 | 41.70 | 70.50 | 72.20 | 38.60 | 38.60 |
| **UA (%)** | 65.10 | 73.70 | 74.10 | 92.90 | 84.50 | 73.10 | 69.90 |
| **OA (%)** | 75.80 | 73.50 | 73.10 | 87.00 | 84.90 | 72.10 | 71.20 |
| **OE (%)** | 26.10 | 56.27 | 58.31 | 29.49 | 27.80 | 61.36 | 61.36 |
| **CE (%)** | 34.93 | 26.29 | 25.90 | 7.14 | 15.48 | 26.92 | 30.06 |
| **Kappa** | 0.49 | 0.38 | 0.36 | 0.71 | 0.66 | 0.34 | 0.32 |
| **Site 4** | | | | | | | |
| | NDWI | MNDWI1 | MNDWI2 | AWEIsh | AWEInsh | MuWI_C | MuWI_R |
| **PA (%)** | 22.40 | 36.80 | 35.10 | 54.60 | 67.20 | 52.30 | 54.00 |
| **UA (%)** | 53.40 | 86.50 | 85.90 | 87.20 | 75.50 | 78.40 | 85.50 |
| **OA (%)** | 78.90 | 85.00 | 84.60 | 88.40 | 88.10 | 86.50 | 88.00 |
| **OE (%)** | 77.59 | 63.22 | 64.94 | 45.40 | 32.76 | 47.70 | 45.98 |
| **CE (%)** | 46.58 | 13.51 | 14.08 | 12.84 | 24.52 | 21.55 | 14.55 |
| **Kappa** | 0.21 | 0.44 | 0.43 | 0.61 | 0.64 | 0.55 | 0.59 |

Note: The omission error was higher than the commission error, irrespective of water index and site.

On the other hand, AWEI was more effective than all other indices at extracting water bodies from all the studied sites. AWEIsh and AWEInsh extracted discontinuous portions of narrow rivers in sites 2, 3, and 4 (Figures S2–S4) and were more efficient at outlining the boundaries of lakes including narrow craters (approximate areas ranging between 0.06–59.33 ha in scene 2, Table S8) that have densely vegetated slopes (Figures 4 and 5).

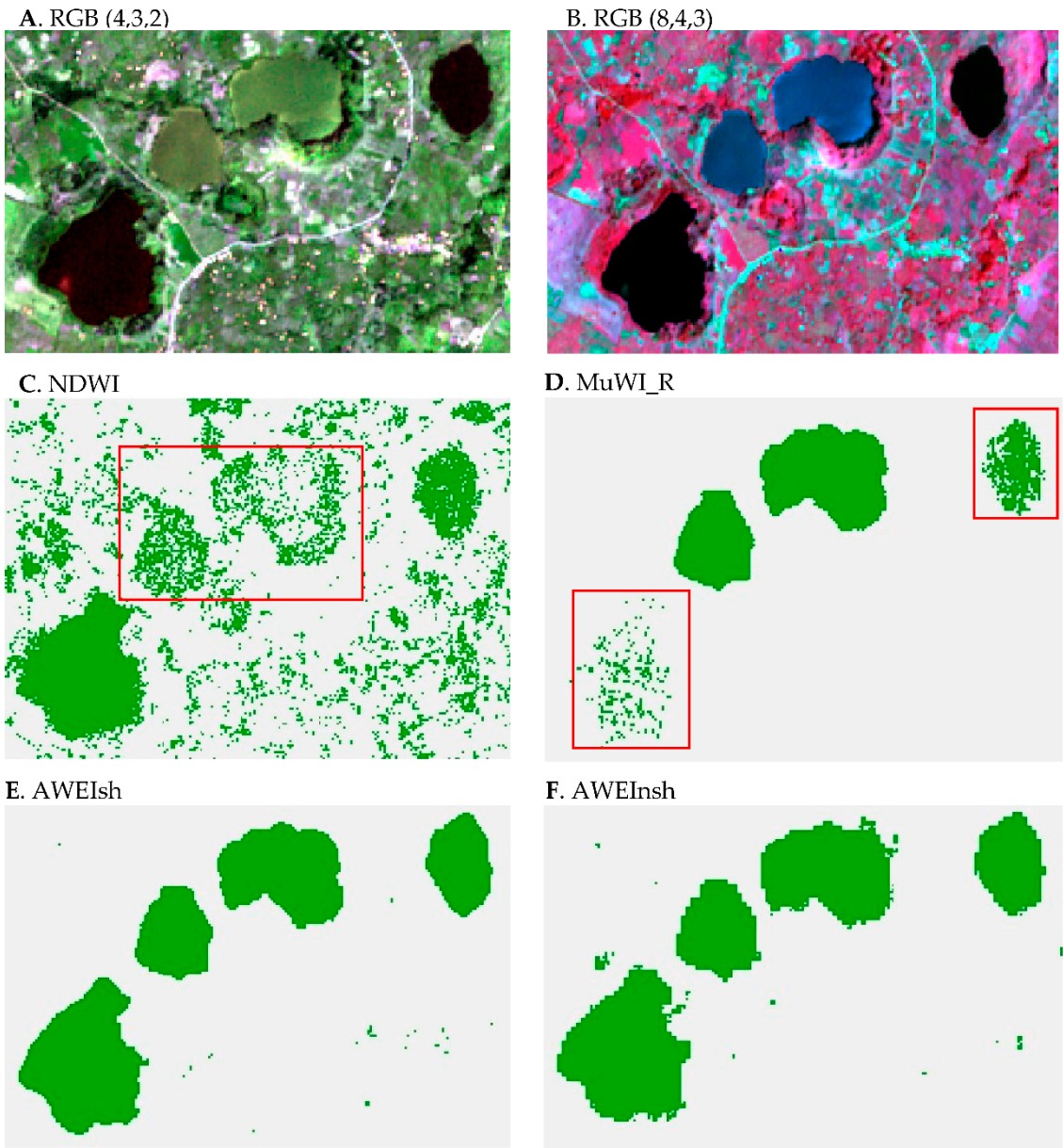

**Figure 4.** Comparing cases of crater lake extraction from site 2. Frame (**A**) is Sentinel-2 MSI RGB (4,3,2) 10 m natural-color image, Frame (**B**) is Sentinel-2 MSI RGB (8,4,3) false-color composite. In frames (**C–F**), green is water while grey is non-water. Frame (**C**) shows results of NDWI where some crater lakes were omitted (red rectangle) and background noise is misclassified as water, (**D**) is MuWI_R which misclassified two crater lakes (red rectangles) but effectively eliminated the background noise, (**E**,**F**) are AWEIsh and AWEInsh, respectively which effectively extracted the crater lakes and eliminated non-water background noise.

In general, the best visual results of surface water bodies extraction in this study as compared to high-resolution images (3 m) available from Google Earth Pro™ and expert's knowledge, were given by AWEI as illustrated in Figure 6. Detailed illustrations of the qualitative analysis for all the water indices in this study are shown in Figures S1–S4 of the Supplementary Materials.

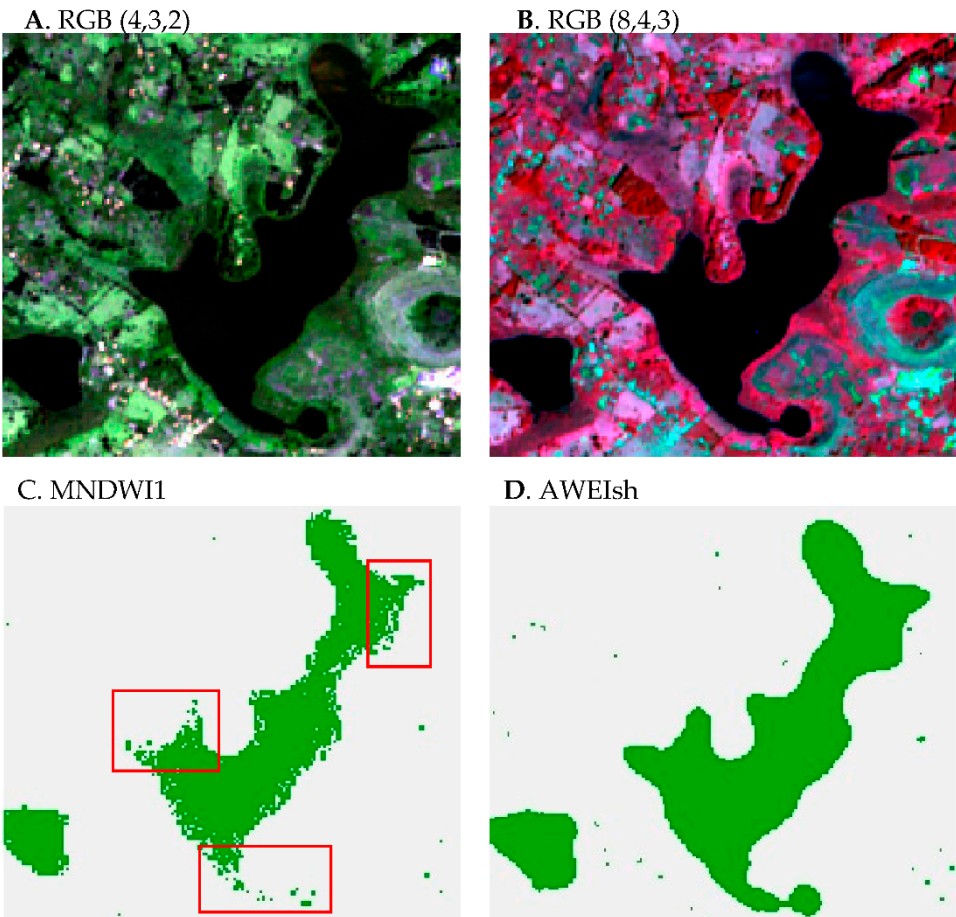

**Figure 5.** Comparing a case of lake extraction in site 1. Frame (**A**) is Sentinel-2 MSI RGB (4,3,2) 10 m natural-color image, Frame (**B**) is Sentinel-2 MSI RGB (8,4,3) false-color composite. Frame (**C**) shows MNDWI1 extraction of Lake Saka with omitted portions (red rectangles), while Frame (**D**) is a more efficient extraction of the same lake using AWEIsh.

*3.2. Quantitative Assessment*

Using the validation points (Figure S6), accuracy assessment was derived following the procedures described in Section 2.6. The confusion matrices for all water indices and sites are in Table S7 of the Supplemental Materials.

Generally, the PA was lower than UA with the only exception being observed in the results of NDWI (site 3), where PA (73.90%) was higher than UA (65.10%). Apart from the observations in site 3 where NDWI (73.90%) had the highest PA, AWEI consistently showed the highest PA in all the other sites. UA was higher than 90% for all water indices in site 1. While in site 2, only MNDWI2 had UA (87.20%) which was less than 90%.

In site 3, AWEIsh and AWEInsh had the highest UA of 92.90% and 84.50% respectively, while the remaining water indices had UA less than 80.00%. On the other hand, NDWI had the lowest UA (53.40%) in site 4, while all the other water indices had UA greater than 75.50% (observed in the results of AWEInsh, Table 3).

AWEI consistently showed the highest OA as compared to all the other water indices, irrespective of the site. OA was greater than 90.00% for AWEI in sites 1 and 2, while it was higher than 87.00% for sites 3 and 4 with the only exception being observed in the results of site 4 of AWEInsh (OA = 84.90%, Table 3).

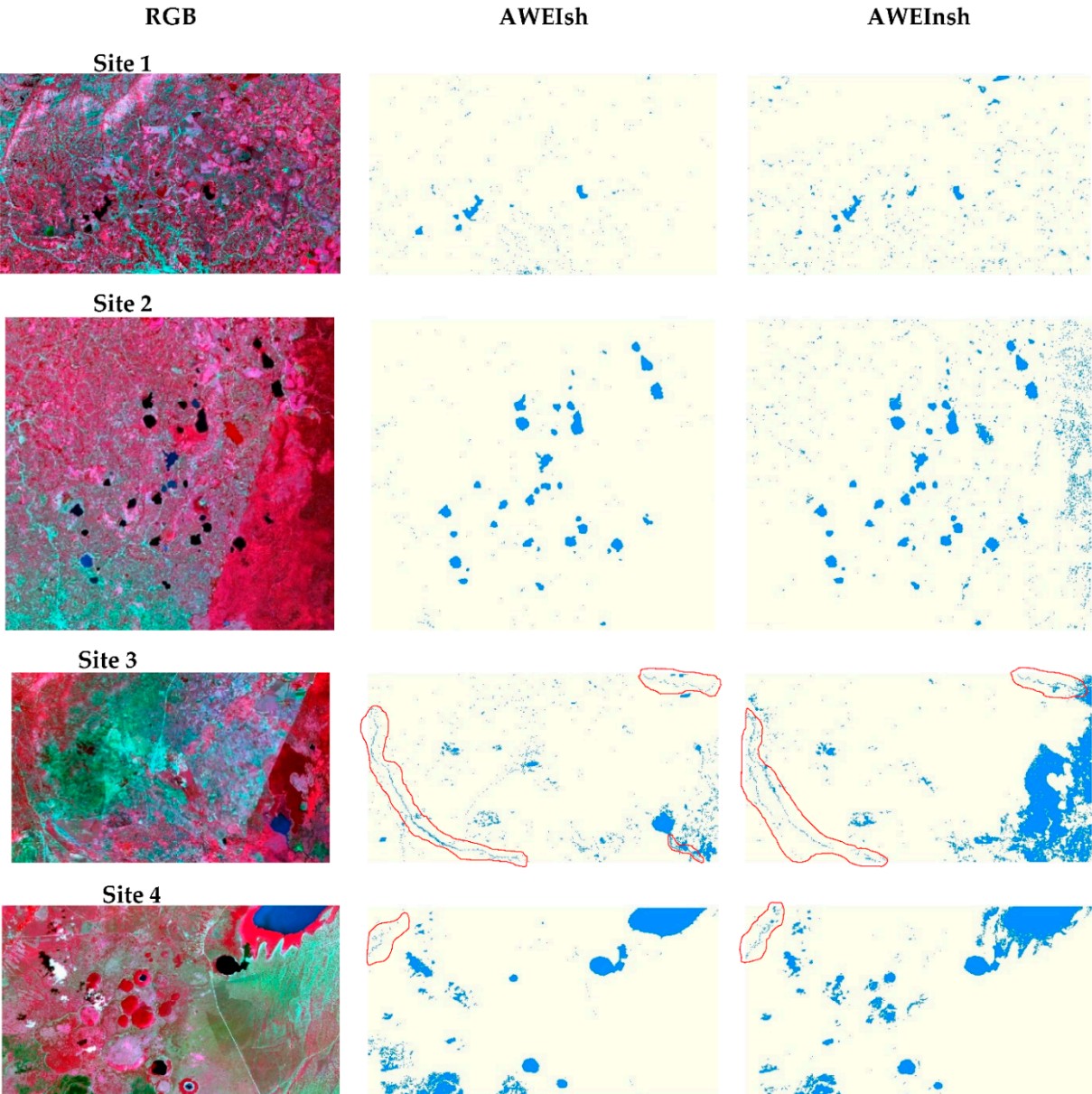

**Figure 6.** Water maps as classified by AWEI for the various sites (Blue is water while white is non-water). The red highlights show rivers or streams. Comparisons follow rows where the Sentinel-2 MSI RGB (8,4,3) false-color composite image of a particular site is on the (**left**), followed by the AWEIsh binary classified image (**middle**), and then the AWEInsh binary classified image (**right**). Further illustrations of water maps are in Figures S1–S4.

The highest kappa was also observed in AWEI, irrespective of the site. The kappa was greater than 0.80 for AWEI in sites 1 and 2, while it was less than 0.80 for all the other five water indices in the same sites. In site 3, AWEIsh and AWEInsh had kappa of 0.71 and 0.66 respectively, while the remaining water indices had less than 0.49 (observed in the results of NDWI). AWEIsh and AWEInsh had kappa of 0.61 and 0.66 respectively in site 4, while the remaining water indices had less than 0.59 (observed in the results of MuWI_R, Table 3).

CE was generally less than the OE irrespective of the water index, for all the studied sites. In sites 1, 2, and 4, NDWI showed the highest OE of 56.02%, 69.52%, and 77.59%, respectively, while MuWI_C and MuWI_R had the highest OE (61.36%) in site 3. AWEI consistently showed the lowest OE in all the studied sites (Table 3).

In sites 1 and 2, MuWI_C showed the lowest CE (0.63% and 0.68%, respectively). On the other hand, AWEIsh showed the lowest CE in sites 3 and 4 of 7.14% and 12.84%,

respectively. The trend observed in the results presented in Table 3 shows that extraction of water using these water indices was more accurate in sites 1 and 2 as compared to sites 3 and 4.

## 4. Discussion

Water resources are vital in the process of eradicating poverty and socio-economic development [1]. Though a lot of information is available about the major lakes and rivers of Uganda [1,60], understanding the minor water bodies that play a pivotal role in the survival of most rural communities of the country is not prioritized. Despite being strategically located in two of the most important drainage sub-basins of Lake Edward and Lake Albert [1], the Rwenzori region exhibits high malnutrition levels among its population [5–7]. Aquaculture could be a sustainable remedy to this problem through the provision of protein, micronutrients, and essential fatty acids [61,62] to the vulnerable populations in this region. Availability of adequate, year-round surface water bodies is one of the most important requirements for aquaculture development [63,64]. Therefore, this study focused on assessing the efficiency of various water indices to map the available inland water resources (streams, rivers, crater lakes, and lakes) for potential inland freshwater aquaculture in the Rwenzori region. Results of seven water indices (NDWI, MNDWI1, MNDWI2, AWEIsh, AWEInsh, MuWI_C, and MuWI_R) over four sites were compared.

AWEIsh and AWEInsh were the most efficient indices at extracting mixed water pixels and produced more visually clear water maps since they eliminated most classification errors of hilly shadows and other non-water surfaces similar to what was observed by Acharya et al. (2018) [43]. Though AWEInsh is meant for areas that are not affected by shadows [16], it was still efficient at extracting surface water in all the studied sites. This was because the region possesses several dispersed built-up areas as background noise which this water index effectively eliminated as discussed by Feyisa et al. (2014) [16]. Its other variant AWEIsh also successfully removed the shadows that AWEInsh could not eliminate as discussed by Feyisa et al. (2014) [16]. The topographic characteristics of the Rwenzori region could have contributed to AWEI performing the best in all incidences (OA > 90%, kappa > 0.8 in sites 1 and 2; OA > 84.9%, kappa > 0.61 in sites 3 and 4, Table 3) because it has previously been commended for its effectiveness at extracting water with high accuracy in mountainous regions where deep shadows of the terrain cause error [16]. On the other hand, NDWI could not eliminate noise from built-up areas (Figures S1–S4) because the reflectance pattern of built-up land in the green and NIR bands is similar to that of water [44]. This water index had the highest omission errors in scenes 1, 2, and 4 where many mixed pixels of narrow and small water bodies were left out, concurring with the observations of Acharya et al. (2018) [43]. Similar to the findings of Acharya et al. (2018) [43], the commission errors of NDWI resulted from misclassifying hilly shadows as water. Acharya et al. (2018) [43] further observed that subzero optimum thresholds of NDWI like the ones obtained in this study resulted in the wrong classification of shadows as water. This observation is further supported by the findings of Wang et al. (2018) [15], who noted that NDWI was greatly affected by commission errors despite using two native 10-m bands on Sentinel-2.

MNDWI replaced the NIR band in NDWI with SWIR to reduce commission errors from built-up areas and bare-soil [15,43] but both versions of this index were still prone to misclassifications of such a kind in all the studied sites just as it was observed in the results of Fisher et al. (2015) [65]. Even MuWI that combines more bands with adequate formations that have the potential of improving shadow detection [15] still exhibited high commission errors (>14%) in sites 3 and 4 (Table 3). MuWI also omitted portions or whole crater lakes and other water bodies (Figure 4 and Figures S1–S4). This could be attributed to the different water properties of the various water bodies of this region. Though the variants of this index have been commended for reducing misclassifications resulting from sunglint due to their efficient exploitation of the native 10-m Sentinel-2 spectral bands [15], they still performed worse than AWEI in all sites.

Except for AWEI (AWEIsh and AWEInsh), all other water indices extracted the crater lakes and water bodies less efficiently when visually examined (Figures 4 and 5, and Figures S1–S4). This could be attributed to the fact that classifying water from remotely sensed images is affected by the inconsistency in reflectance spectra of water bodies that have different properties such as color, turbidity, depth, steepness of topography, dissolved organic matter, organic particulates, sediment, and/or plankton concentration [65], which is a sure case for Rwenzori region as illustrated in the discussions of Nankabirwa et al. (2019) [66].

The higher user's accuracy than producer's accuracy for all water indices in this study, irrespective of either water index or scene (Table 3), solidifies the fact that there was far less misclassification of non-water pixels as water than the misclassification of water pixels as non-water. Therefore, the quantitative results of this study, just like what was obtained by Fisher et al. (2015) [65] and Feyisa et al. (2014) [16], showed that the omission error was higher than the commission error. The high omission error resulted from the fact that many mixed water pixels and narrow water bodies could not be effectively extracted from Sentinel-2 L2A imagery. Most water bodies (streams, and craters) in the Rwenzori region are characterized by being narrow and having densely vegetated (forested) banks. Such characteristics, coupled with the rugged terrain of the region (characterized by high and steep slopes) led to the high omission errors. Lara et al. (2013) [49] noted that the streams draining the Rwenzori mountains are generally small and of low stream order. This is the reason why they could not be mapped using Sentinel-2 imagery.

Though the omitted streams are narrow, the majority are permanent due to being fed by the very high rainfall levels experienced in this region [1,60,67] and the melting glaciers of the Rwenzori mountains [67]. Several recent studies have indicated that most of the streams and rivers of this region have good to excellent water quality [8–10], which is a white flag to their high potential for use in aquaculture production. Streams and rivers are in general a highlighted major source of water for aquaculture [68,69], which is also the case for Uganda where they have been identified as vital to pond fish farming [70–72]. Results from a previous study [71] showed that the majority of the active fish farmers in the Rwenzori region used streams as their main source of water for their fish ponds. Therefore, the failure of water indices to extract narrow streams from Sentinel-2 L2A satellite imagery leads to a severe underestimation of the potential water resources for aquaculture using remote sensing.

On the contrary, several freshwater crater lakes in the region were successfully extracted by AWEI (Figure 6). For example, 39 successfully identified crater lakes with surface area ranging between 0.06 and 59.33 ha were extracted by these indices in site 2 (Table S8). Lake Saka (surface area = 56.90 ha; maximum depth = 11.9 m [66]) was also one of the lakes extracted from site 1 (Figure 5). This lake possesses an active artisanal fishery (A. Ssekyanzi, personal observation) which is proof that it could potentially be used for aquaculture production. The importance of such water resources to the survival of the rural communities of the Rwenzori region has been mentioned in previous studies [49,66].

Crater lakes are being utilized for aquaculture production in various parts of the world [73,74]. A previous limnological study [66] showed that some of the crater lakes that were identified in this exercise have good water quality parameters for aquaculture development such as depth, temperature, turbidity, dissolved oxygen, pH, total phosphorus, total nitrogen, and primary productivity. Therefore, these could be potential sites for cage and/or pen culture, which is not yet popular in this region of Uganda.

## 5. Conclusions

Of the water indices compared in this study, AWEI showed the highest accuracy (OA and kappa) in mapping water bodies from the studied sites. The findings of this study showed that the Rwenzori region possesses surface water resources such as crater lakes, lakes, and rivers/streams that could potentially support aquaculture development.

However, the high omission errors observed are an indication that the majority of the lower order streams that characterize the water resources of the region could not be identified by applying water indices on moderate resolution Sentinel-2a/b images. Since spatial analysis alone cannot unravel the dynamics and seasonal variations of these water resources, it is necessary to carry out a spatial-temporal analysis of the surface water resources over a given period.

The results of this paper are a contribution toward the unearthing of the aquaculture potential of the Rwenzori region.

**Supplementary Materials:** The following are available online at https://www.mdpi.com/article/10.3390/w13192657/s1. Figure S1: RGB images and the corresponding water maps of scene 1. Figure S2: RGB images and the corresponding water maps of scene 2. Figure S3: RGB images, and the corresponding water maps of scene 3. Figure S4: RGB images and the corresponding water maps of scene 4. Figure S6: RGB images of the studied scenes (1, 2, 3, and 4) with the reference points (red) used for the accuracy analysis. Table S5: K-means cluster centers, where the optimum threshold (the center that gave the highest OA and kappa) is highlighted. Table S7: Confusion matrices of the water indices under the respective studied scenes of the Rwenzori region. Table S8: 39 extracted crater lakes by AWEI in scene 2 with their approximate area (m$^2$/ha).

**Author Contributions:** Conceptualization, A.S., N.N., and G.V.S.; methodology, A.S. and D.V.d.Z.; validation, A.S.; formal analysis, A.S.; investigation, A.S.; data curation, A.S. and M.E.B.; writing—original draft preparation, A.S.; writing—review and editing, A.S., M.E.B., N.N., G.V.S., and D.V.d.Z.; visualization, A.S.; supervision, N.N. and G.V.S.; project administration, G.V.S.; funding acquisition, N.N. and G.V.S. All authors have read and agreed to the published version of the manuscript.

**Funding:** This research was funded by VLIR-UOS (funding agency) for DGD (Belgian Government), grant number UG2019IUC027A103.

**Institutional Review Board Statement:** Not applicable.

**Informed Consent Statement:** Not applicable.

**Acknowledgments:** Our gratitude goes to the VLI-ROUS for the scholarship that enabled us to carry out this study.

**Conflicts of Interest:** The authors declare no conflict of interest.

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
