# Peer review of "Identification of Potential Surface Water Resources for Inland Aquaculture from Sentinel-2 Images of the Rwenzori Region of Uganda"

_water, doi:10.3390/w13192657_

Round 1
Reviewer 1 Report
The manuscript by Ssekyanzi et al entitled "Identification of potential surface water resources for inland aquaculture from Sentinel-2 images of the Rwenzori region of Uganda" investigates the use of Sentinel-2 imagery to map inland water bodies in the Rwenzori region of Uganda using seven spectral indices of relevance to mapping water.
The paper is generally well written and of potential interest to Water's readership. The authors describe the methods used clearly , appropriately discussing the main findings and supporting the conclusions with the results obtained. For these reasons I recommend accepting the paper for publication after the following minor revision.
General comments
I would recommend to discuss how would the next generation of satellites, specifically designed for surface water analysis (i.e. SWOT), improve water availability estimates in Rwenzori region. Additionally, it would be appropriate to discuss the approaches recently used to develop global surface water data sets using imagery from satellites with different spatial and temporal resolutions (e.g. Landsat, MODIS) and their applicability to the Rwenzori region specifically for the goals of this study. See examples below.
Pekel et al., 2016 (doi: 10.1038/nature20584) and references therein.
Tortini et al., 2020 (doi: 10.5194/essd-12-1141-2020) and references therein.
I would also suggest to further discuss the limitations of the methods used, and the significance and applicability of the findings to map water bodies in other regions worldwide.
Specific comments
L15. Please avoid using qualitative adjectives such as "huge".
L22. If "best", consequently also "only".
L29. Please better explain what the authors mean with "invisible". To other remote sensing platforms?
L37-38. Please provide reference for "Sustainable Development Goal 2".
L39. Quantify "immense".
L42. Please explain why "ironically".
L44. See L29 comment.
L45. Please cite global water mapping studies (see references in general comments) and not only local/regional studies. It would be of interest to separate these studies by scale and discuss their significance for the Rwenzori region.
L81. Avoid "applauded".
L116. Use either "for" or "in", not "during".
L119. How dynamic ("varying sizes") are the water bodes investigated? Have the authors considered replicating the study in dry/wet conditions?
L131. Why did the authors use just one image and not the entire Sentinel-2 stack available? Contemporary remote sensing case studies have been shying away from using a single snapshot approach.
L132. Is the "Sentinel level-2A product" a Sentinel-2 L2A image? Or else?
L147-148. The terminology used is at times confusing. A "scene" should refer to the image, not the clips used. Please address throughout the manuscript.
Table 1. Round bracket missing in the NDWI equation.
L172. Why "for AWEI" in brackets?
L185-186. How was the noise level evaluated? And what do the authors mean with "confusing pixels"? Based on the explanation provided, land-water boundary pixels might in fact be representative of water based on the dynamics of the water bodies studied. Please provide more details.
L190-202. Confusion matrix unnecessary here, an appropriate reference might suffice.
L210. Did the authors precisely orthorectify the four clips before their qualitative evaluation?
L227. Consider using "outlining" instead of "demarcating".
Figure 4 and 5. I would recommend to add the definition of the product visualized in each panel (e.g. RGB, NDWI, etc).
L241. How were the "visual results" evaluated?
L295. Why "tremendously"?
L296-298. Reference needed.
L305-308. Have the authors considered using an object- rather than a pixel-based approach? If so, please explain why such approach was not deemed appropriate. If not, please discuss pros and cons of both with reference to the goals of the study.
Author Response
Dear Reviewer,
We are grateful for your contribution to our work. Attached is the revised version of the manuscript and responses to your comments.
Athanasius Ssekyanzi (On behalf of authors).

Reviewer 2 Report
This paper’s approach is based on [33] and its aim is to compare the performance of several spectral indices derived from one Sentinel-2 scene to accurately detect water/non-water regions. In addition, it is the first study of this kind for Uganda’s which is important as this region has ubiquitous and variable water resources. The authors also take [33] one step further by defining the threshold value between water/non-water using k-means cluster analysis iteratively instead of doing this manually. The optimal k-means parameters are obtained based on the Overall Accuracy an kappa coefficient. They perform the accuracy assessment by creating labeled pixels of the area based on expert’s Google Earth photo-interpretation.
Broad Comments (Abbreviations: Line Number: LN; Figure: FIG; Table: TAB)
- Generally, the manuscript is well-written and structured. The authors have presented adequate bibliography for the topic, although more information and background could have been mentioned in the case of cluster analysis and the atmospheric correction step.
- In addition, the authors should pay more attention to the visualizations since they could have been more self-explanatory.
- TAB2 could be moved to the supplementary material.
- Whether to apply atmospheric correction (AC) and corresponding type seems to be not such straightforward as you mention in LN134-136, and especially for the case of water detection. Despite it seems that generally in the classification task of a single image AC might not have such an important impact, when it is applied for water detection there are a couple of parameters that may need more attention. For example, issues such as sun glint and adjacency effect might need attention. Therefore, I would suggest you explain a little bit more about the specific AC that Sentinel-2 Level-2A products come with, whether sun glint/adjacency effect is important or not for your study, as well as whether some water indices (e.g. heavy normalized differences) you use may counterbalance the potential need for different ACs.
- How do you guarantee that the labeled classes (water/no-water) are not heavily imbalanced and that the resulting accuracy assessment is not heavily biased due to this?
Specific Comments
- LN21 Change “Sentinel 2a/b” to “Sentinel-2A/B”
- LN23 Change “overall accuracies” to “Overall Accuracy (OA)”
- LN88-91 Maybe the additional study about threshold should go at the end of the paragraph that starts on LN65
- LN141 Which resampling algorithm was used?
- LN167-169 Can you expand this a little bit? You could mention the number of k (I see that it’s 2). By “color range of water in the index plots” you mean the cluster that represents the water? How many iterations (trial and error) did you run?
- LN183 Maybe change from “labeled” to “manually labeled” to make it 100% clear
- LN185 “comprised” instead of “were comprised of”
- LN179-189 Please mention what is the percentage of each class (water, non-water) that you use for the 800 generated points
- LN198-202 Please mention what is the definition of each metric
- FIGS1 There is no green rectangle in the figure
- FIGS1-FIGS4 I think it is not easy enough to examine the figures due to lack of a legend. Alternatively, the pixels misclassified as water could have been annotated with a different color depending on their true class. In this way the comparison between the results of the spectral indices would be easier. However, I see the point.
- LN214-215 Isn’t it expected that your algorithms cannot detect narrow water bodies since this kind of detection (based solely on spectral indices) problem is bounded by sensor’s spatial resolution which is not high enough and the e.g. narrow rivers signal is lost (or mixed with adjacent materials) ?
- LN258-259 PA of AWEI for scene 3 comes as the second highest, though
- LN335 There is no MuWI in FIG5
- LN341 I think “color” can be omitted since it’s the result of the parameters you already mention (e.g. plankton concentration, sediment etc.)
- LN346 “either … or…” instead of “neither … nor...”
Author Response

(The authors gave the same response as above.)

Reviewer 3 Report
The paper provides many adequate citations and is clearly written. Please address the following comments below prior to publication.
line 15: Indicate that “animal protein” is a commercial demand, and indicate the associated industries.
lines 16-17: What is meant by “limited knowledge”? Is this the requisite scientific knowledge required to make decisions?
line 22: Define the acronyms “AWEIsh” and “AWEInsh” in the abstract of the paper as the indices that are later referenced in the paper.
line 23: Define “kappa coefficient” for the abstract.
Abstract: Indicate why the water indices were not able to extract streams. Is this due to image resolution?
line 25: Why is there an opportunity? How is this opportunity implemented within the social geography of the country?
line 40: Consider using a term other than “stunted.” There are more descriptive terms related to medical disorders.
lines 43: What are the sources of “good quality” water, and what is meant by “good quality”? Can this be expressed as water quality, or described as potable water?
lines 76-77: What is meant by “complete version”?
line 183: How were the images labelled? Is this a type of training dataset?
line 292: What is the “minimal effort”?
Section 4: The discussion does not have to repeat information included earlier in the document.
line 385: Why is band resampling thought to improve the accuracy of water indices? Provide a reference or a discussion here.
Conclusion: Indicate here how DEM models can be merged with the remote sensing images to obtain a better indication of lower order streams.
Author Response
Dear Reviewer,
We are grateful for your contribution to our work. Attached is the revised version of the manuscript and responses to your comments.
Athanasius Ssekyanzi (On behalf of authors).

This manuscript is a resubmission of an earlier submission. The following is a list of the peer review reports and author responses from that submission.
Round 1
Reviewer 1 Report
Dear authors,
I enjoyed reading/reviewing this paper but some sections (2.6 & 3.2) assessment in the manuscript needs to be rewritten in order to have a clear message. also I think there is some error in figure assignment. my comments attached.
Regards.

Author Response
Dear Reviewer,
Thank you for reviewing our work. Your comments were mind-opening towards improving the quality of this paper. We have incorporated all the suggestions mentioned in your comments and modified sections 2.6 and 3.2 accordingly, in the revised version.
Thanks a lot!
Athanasius Ssekyanzi.
Reviewer 2 Report
The study is significant for environmental concerns and economic and infrastructure development. It is more important for a developing country where this kind of study has never been performed.
Although, the objective of this study is of high importance, the study is basically an application of existing tools and methods.
Here are my comments:
- Regarding temporal analysis: Source of water is not static. It contracts and expands seasonally. The long term trend in this contractions and expansion provides critical signals of drying or wetting. Hence it would be meaningful to obtain temporally optimized index threshold. Temporal consistency would be another criteria to measure the performance of an index method and would provide uncertainty analysis for a method and in its prediction. This may also help to identify the relationship of bands with the changing index thresholds.
- Regarding spatial analysis: This study used various index methods to identify water bodies. Each index method is based on a critical threshold value to separate other objects from water. The authors performed clustering analysis to obtain a threshold of the index. This study may include the transfer of a threshold value identified at one location to another and evaluate the consistency performance of an index method.
- Include actual water body boundary. For example: including actual water body boundaries in Figure 4 would better represent the quality of model predictions.
Author Response
Dear Reviewer,
Thank you for reviewing our work. Your comments were mind-opening and were helpful towards improving the quality of the paper.
Please see the attachment of responses to your comments.
Athanasius Ssekyanzi.

Reviewer 3 Report
The classification of the surface water areas is an important work in the remote sensing image applicaiton. It is easier to get high accuracy than other land use and land cover types. This manscript used serven remote sensing derived water indexes to extract the water areas from Sentinel 2a/b images. The K-means cluster was used to claasified the water area. I think that the methods are not novel enough to justify publication. From the Table 4, it is indicated that the classification accuracies are also very low based on the relative simple claasifer. And the manuscript is also not clear enough for publication as the current version.
Author Response
Dear Reviewer,
Thank you for reviewing our work. Your comments were mind-opening towards improving the quality of this paper. Please see the attachment of the responses to your comments.
Athanasius Ssekyanzi

Round 2
Reviewer 2 Report
This study is significant in a developing country like Uganda.
Although the study is limited to spatial analysis of water indices, this reviewer accepts the manuscript to be published in the Journal Remote Sensing. I hope the authors will include the temporal analysis of the water indices in their further study. Figure 4A is missing. Supplementary material does not include table S9.
Reviewer 3 Report
Even though the authors try their best to improve the manuscript, I still think that the manuscrip is lack of novelty. The second reviwer gave the similar view on the innovation with me. I suggest that the author can submit the manscript to the Water.